# Diet and Exercise Modulate GH-IGFs Axis, Proteolytic Markers and Myogenic Regulatory Factors in Juveniles of Gilthead Sea Bream (*Sparus aurata*)

**DOI:** 10.3390/ani11082182

**Published:** 2021-07-23

**Authors:** Miquel Perelló-Amorós, Isabel García-Pérez, Albert Sánchez-Moya, Arnau Innamorati, Emilio J. Vélez, Isabel Achaerandio, Montserrat Pujolà, Josep Calduch-Giner, Jaume Pérez-Sánchez, Jaume Fernández-Borràs, Josefina Blasco, Joaquim Gutiérrez

**Affiliations:** 1Departament de Biologia Cellular, Fisiologia i Immunologia, Facultat de Biologia, Universitat de Barcelona, 08028 Barcelona, Spain; miquelperelloamoros@gmail.com (M.P.-A.); isabelgarcia@ub.edu (I.G.-P.); alsanchezmo@ub.edu (A.S.-M.); arnauic98@gmail.com (A.I.); jaume.fernandez@ub.edu (J.F.-B.); jblasco@ub.edu (J.B.); 2Université de Pau et des Pays de l’Adour, E2S UPPA, INRAE, UMR1419 Nutrition Métabolisme et Aquaculture, F-64310 Saint-Pée-sur-Nivelle, France; emilio-jose.velez-velazquez@inrae.fr; 3Department d’Enginyeria Agroalimentària i Biotecnologia, Escola Superior d’Agricultura de Barcelona, Universitat Politècnica de Catalunya BarcelonaTech, 08860 Barcelona, Spain; maria.isabel.achaerandio@upc.edu (I.A.); montserrat.pujola@upc.edu (M.P.); 4Nutrigenomics and Fish Growth Endocrinology, Institute of Aquaculture Torre de la Sal (CSIC), Ribera de Cabanes, 12595 Castellón, Spain; j.calduch@csic.es (J.C.-G.); jaime.perez.sanchez@csic.es (J.P.-S.)

**Keywords:** aerobic training, muscle remodeling, endocrine regulation, hypercaloric-diets, fillet quality

## Abstract

**Simple Summary:**

The effects of exercise and diet on growth markers were analyzed in gilthead sea bream juveniles. Under voluntary swimming, fish fed with a high-lipid diet showed lower growth, growth hormone (GH) plasma levels, flesh texture, and higher expression of main muscle proteolytic markers than those fed with a high-protein diet. However, under sustained exercise, most of the differences disappeared and fish growth was similar regardless of the diet, suggesting that exercise improves nutrients use allowing a reduction of the dietary protein, which results in an enhanced aquaculture production.

**Abstract:**

The physiological and endocrine benefits of sustained exercise in fish were largely demonstrated, and this work examines how the swimming activity can modify the effects of two diets (high-protein, HP: 54% proteins, 15% lipids; high-energy, HE: 50% proteins, 20% lipids) on different growth performance markers in gilthead sea bream juveniles. After 6 weeks of experimentation, fish under voluntary swimming and fed with HP showed significantly higher circulating growth hormone (GH) levels and plasma GH/insulin-like growth-1 (IGF-1) ratio than fish fed with HE, but under exercise, differences disappeared. The transcriptional profile of the GH-IGFs axis molecules and myogenic regulatory factors in liver and muscle was barely affected by diet and swimming conditions. Under voluntary swimming, fish fed with HE showed significantly increased mRNA levels of *capn1*, *capn2*, *capn3*, *capns1a*, *n3,* and *ub*, decreased gene and protein expression of Ctsl and Mafbx and lower muscle texture than fish fed with HP. When fish were exposed to sustained exercise, diet-induced differences in proteases’ expression and muscle texture almost disappeared. Overall, these results suggest that exercise might be a useful tool to minimize nutrient imbalances and that proteolytic genes could be good markers of the culture conditions and dietary treatments in fish.

## 1. Introduction

One of humanity’s greatest challenges is feeding a constantly growing population, in a situation in which the availability of natural resources is limited and respect for ecosystems must be a priority. In this context, the aquaculture sector can help face this demand by providing aquatic products like fish, one of the healthiest sources of high-quality protein, fat, vitamins, and oligo elements [1]. However, there is still a need to develop the sector towards a better sustainability combined with the improvement of fish growth and product quality. In this sense, the reduction of dietary protein was a priority in many nutritional studies since it is the most expensive component in aquafeeds, its catabolism is the main source of water nitrogen loading [2,3,4,5], and the ecological impact (overfishing and habitats destruction) that the use of fish components accounts. The maintenance and handling of fish during rearing is another area in which improvements can be made, and in this regard, the effects of exercise as a tool to increase growth rate and flesh quality were widely studied in different species [4,6,7,8,9,10,11,12,13,14,15,16]. Moderate and sustained swimming results in clear benefits in total fish growth, muscle structure, and both metabolic and endocrine status [4,6,9,10,11,12,13,14]. Interestingly, exercise changes nutrient requirements and utilization, and this activity may increase the inclusion of more sustainable nutrients in the diet, such as lipids or carbohydrates, which allow the reduction of nitrogen discharges and feeding costs. In fact, this study is an extension of a previous work which studied how swimming activity modulate the effects of different dietary protein/lipid ratios on growth rate, tissue composition, and energy metabolism in the same experimental fish [5]. In that study, it was reported that the lower protein/lipid ratio affected growth through unbalanced availability and use of nutrients, especially indicated by changes in key mitochondrial proteins related to energy metabolism; however, the sustained exercise counteracted most of these alterations.

Somatic growth in vertebrates is mainly regulated by the hypothalamic-pituitary axis through the growth hormone (GH) and the insulin-like growth factors (IGFs) system [17], and many authors reviewed its role in fish species [18,19,20,21]. In gilthead sea bream, changes in the GH-IGFs axis were studied through the seasonal cycle [22], life stages [23], nutritional status [24,25], as a response to environmental factors [26,27], or even after several weeks of sustained exercise [14,28]. All these works highlighted the important regulatory functions of those endocrine factors in this species. Once GH is secreted by the anterior pituitary gland, it is transported via the blood, reaching the target tissues in which it interacts with the corresponding membrane receptors (GHR-1 and GHR-2) to trigger its effects. Differential regulation of the GH-IGFs axis components by nutrients and environmental factors was reported at a transcriptional level in fish liver and muscle (reviewed by Pérez–Sánchez et al. [29]). In liver, GH stimulates the production and release of IGFs, which are subsequently sequestered by the IGF-binding proteins (IGFBPs) to modulate their distribution and bioavailability [20]. In gilthead sea bream, Tiago et al. [30] identified three splice variants of *igf-1* (*igf-1a*, *igf-1b*, and *igf-1c*) that were differentially expressed in tissues. They observed that *igf-1c* was the most expressed isoform in the hepatic tissue, suggesting an important systemic role for this splice variant. IGFs exert their effects through the IGF-1 receptor(IGF-1R) and the IFG-2 receptor (IGF-2R), which should be also considered to better understand how all these factors are implicated in growth control.

Besides the GH-IGFs axis, other important molecules are involved in fish muscle development, such as the proteolytic systems and the myogenic regulatory factors (MRFs) that contribute sequentially directing muscle fibers recycling and formation [31,32]. Proteolytic systems participate in protein degradation and amino acids recycling or catabolism [33,34], and are considered key regulatory factors controlling growth potential with a greater importance during periods of intensive growth (i.e., fingerlings vs. juveniles or adult fish) [13,35]. The main proteolytic systems include the calpains, the lysosomal cathepsins, and the ubiquitin-proteasome (UbP) system [32]. These proteolytic markers were characterized in gilthead sea bream and pointed out as useful molecules to detect muscle remodeling episodes that can affect muscle structure, tenderness, and flesh quality [13,35,36,37]. Moreover, they are essential, especially during exercise, since physical activity seems to affect protein turnover in skeletal muscle to restructure the myofilaments and prevent exercise-induced muscle damage, thus facilitating somatic growth [13]. Furthermore, the exercise was shown to have different effects on the anterior and caudal muscle regions, reflecting the progress of remodeling through the muscle trunk [6,13].

Moreover, the MRFs are involved in muscle hyperplasia (i.e., the recruitment of new fibers) and hypertrophy (i.e., the increase in fiber size), which are processes that occur continuously during the whole fish life, as many species have indeterminate growth [38,39]. Some of these MRFs are crucial to control cell proliferation and muscle lineage determination [myogenic factor 5 (Myf5) and myogenic determination factor (Myod)]; lead to myoblasts fusion and differentiation (myogenin); or are responsible for myotubes maturation [myogenic regulatory factor 4 (Mrf4)] [36,39]. Besides MRFs, myostatin (Mstn) is also important in this myogenic developmental process since it negatively regulates myocytes proliferation and differentiation [40,41].

In this framework, the main objective of this work was to study whether sustained exercise could compensate a low dietary amount of protein balanced with a high lipid composition in juveniles of gilthead sea bream. To this end, changes in the GH-IGFs axis, proteolytic systems and MRFs will be evaluated at transcriptional and protein levels. 

## 2. Materials and Methods

### 2.1. Experimental Design

Nine hundred and eighty gilthead sea breams (4.1 ± 0.1 g body weight) were obtained from a commercial hatchery (Piscimar SL, Burriana, Spain) and reared in the facilities of the Faculty of Biology (University of Barcelona) in a semiclosed recirculation system with a weekly renewal of 20–30%, at 23 ± 1 °C, a salinity of 35–37‰ and a photoperiod of 15 h light/9 h dark. The fish were randomly distributed between two 400 L and eight 200 L tanks at the same biomass density (1.5 kg × m^3^). In the 400 L tanks, fish were kept in voluntary swimming conditions, while in the 200 L tanks, fish were forced to a sustained swimming. To achieve this sustained activity, a tangential laminar flow was created by placing a plastic column in the center of each tank and connecting the water inlet to a vertical tube with lateral holes. The initial flow speed was set at 2.5 body lengths (BL) × s^−1^. Fish were fed with two different commercial diets provided by Skretting Spain SA (Burgos, Spain). Both diets differed mainly in protein and fat content (as illustrated in Table 1) and were named as high-protein (HP: 54% protein/15% lipid) or high-energy diet (HE: 50% protein/20% lipid). Each diet was used to feed half of the tanks, one of 400 L and four of 200 L, thus establishing four experimental conditions. Fish were fed with a ration of 5% of the total biomass of each tank divided in 3 meals per day. The pellet size was increased from 1.5 mm to 1.9 mm after three weeks of experiment, according to the larger size of the fish. Biometric parameters (weight and length) were determined at the beginning of the experiment, after three weeks, and at the end of the trial (6 weeks) for monitoring fish growth. Fish were fasted for 12 h before any manipulation and sampling. As previously reported [5], for the gene and protein expression analysis in the final sampling at week 6, 12 fish from each voluntary swimming tank and 4 fish from each exercise tank (16 fish for each diet) were properly anaesthetized (MS-222, Sigma-Aldrich, Tres-Cantos, Spain), measured, weighed and sacrificed by severing the spinal cord and then eviscerated. Blood from these fish and the fish used for proximal composition [5] was taken from caudal vessels using EDTA-Li (Sigma-Aldrich, Tres-Cantos, Spain) as an anticoagulant and centrifuged (13,000× *g*, 10 min, 4 °C) to separate the plasma, which was stored at −80 °C until further analysis. Samples of liver and both anterior and caudal regions of white muscle were collected and immediately frozen in liquid nitrogen and stored at −80 °C until being processed for the analysis of either gene or protein expression. Moreover, additional 10 fish from the voluntary swimming tanks and 4 fish from the sustained swimming tanks (16 for each diet) were equally sacrificed to obtain a piece of 1.5 × 1.5 cm of white muscle that was extracted from the anterior-dorsal region, bagged, and kept on ice for the evaluation of texture, as explained below. 

All animal-handling procedures were conducted following the guidelines of the Council of the European Union (EU 2010/63), the Spanish and Catalan governments and with the approval of the Ethics and Animal Care Committee of the University of Barcelona (CEEA 663/13 and permit number DAAM 7644).

### 2.2. GH and IGF-1 Plasma Levels

Plasma GH was determined by a homologous gilthead sea bream radioimmunoassay (RIA) as previously described [42]. The sensitivity and midrange (ED50) of the assay were 0.15 and 1.8 ng/mL, respectively. Plasma insulin-like growth factors (IGFs) were extracted by acid–ethanol cryoprecipitation [43], and the concentration of IGF-1 was measured by means of a generic fish IGF-1 RIA validated for Mediterranean perciform fish [44]. The sensitivity and midrange of the assay were 0.05 and 0.7–0.8 ng/mL, respectively.

### 2.3. Gene Expression

#### 2.3.1. RNA Extraction and cDNA Synthesis

To perform the gene expression analysis, tissue homogenization was carried out from 30 or 100 mg of liver or white muscle, respectively. The samples were homogenized in 1 mL of TRI Reagent^®^ Solution (Applied Biosystems, Alcobendas, Spain) using Precellys^®^ Evolution Homogenizer cooled at 4–8 °C with Cryolys^®^ (Bertin Technologies, Montigny-le–Bretonneux, France). After homogenization, RNA extraction was performed following the manufacturer’s TRI Reagent^®^ protocol. The RNA concentration and purity of the samples were determined using the Nanodrop 2200^TM^ (ThermoScientific, Alcobendas, Spain). The RNA integrity was checked in a 1% (*w*/*v*) agarose gel stained with SYBR-Safe^®^ DNA Gel Stain (Life Technologies, Alcobendas, Spain). The RNA samples were stored at −80 °C.

For cDNA synthesis, 1.1 µg of total RNA was treated with DNase I Amplification Grade (Life Technologies, Alcobendas, Spain) and retrotranscribed with the Transcriptor First Strand cDNA Synthesis Kit^®^ (Roche, Sant Cugat del Vallès, Spain). The cDNA obtained was stored at −20 °C until further analysis.

#### 2.3.2. Quantitative Real-Time PCR (qPCR)

The qPCR was carried out following the MIQE guidelines [45] in a CFX384™ Real-Time System (Bio-Rad, El Prat de Llobregat, Spain) using iTAQ Universal SYBR^®^ Green Supermix (Bio Rad, El Prat de Llobregat, Spain) and Hard-Shell^®^ 384-well PCR plates (Bio-Rad, El Prat de Llobregat, Spain). The analyses were carried out in triplicate, using for each well: 2.5 µL of iTAQ Universal SYBR^®^ Green Supermix (Bio–Rad, El Prat de Llobregat, Spain), 1 µL of cDNA, 250 nM (final concentration) of forward and reverse primers and 1.25 µL of DEPC water. The qPCR program consisted of 3 min at 95 °C, 39 × (10 s at 95 °C, 30 s at the melting temperature of the primers and fluorescence detection), followed by an amplicon dissociation analysis from 55 to 95 °C with an increase of 0.5 °C each 30 s.

In the liver, the mRNA transcript levels of total *igf-1*, its three splice variants (*igf-1a*, *igf-1b*, and *igf-1c*), *igf-2*, *igfbp-1a*, *igf-1rb*, *ghr-2*, plus four reference genes (*ef1a*, *rps18*, *rpl27a,* and *tom20*) were analyzed. In the anterior and caudal white skeletal muscle, the gene expression of the same GH-IGFs axis components examined in liver was analyzed, plus *igf-1ra* and *ghr-1*; as well as the expression of proteolytic markers (*ctsl*, *ctsda*, *capn1*, *capn2*, *capn3*, *capns1a*, *capns1b*, *mafbx*, *murf1*, *n3,* and *ub*), MRFs (*myod1*, *myod2*, *mrf4,* and *myogenin*), and growth inhibitors (*mstn1* and *mstn2*), plus four reference genes (*ef1a*, *rps18*, *rpl27a,* and *tom20*). All the primers used in the qPCRs are shown in Appendix A.

Transcript abundance of each studied gene was calculated with the Bio–Rad CFX Manager™ 3.1 software (Hercules, CA, USA) relative to the geometric mean of the combination of the two most stable reference genes (confirmed by the geNorm algorithm), using the method described by Pfaffl [46].

### 2.4. Western Blot Analysis

Protein was extracted from 100 mg of anterior white skeletal muscle in 1 mL of radioimmunoprecipitation assay buffer (RIPA) supplemented with both phosphatases and proteases inhibitors (i.e., PMSF, NA_3_VO_4_ and the cocktail P8340, Santa Cruz, CA, USA) using the Precellys^®^ Evolution Homogenizer cooled with Cryolys^®^. Concentration of soluble protein was determined by the Bradford’s method using bovine serum albumin (BSA, Sigma-Aldrich, Tres Cantos, Spain) for the standard curve. Twenty to forty µg of the soluble protein fraction were prepared in a loading buffer (containing SDS and β-mercaptoethanol), heated at 95 °C for 5 min, and run in a 12% polyacrylamide gel. Following, the proteins were transferred overnight a 4 °C to Immobilon^®^ PVDF-FL 0.2 μm Transfer Membranes (Merck Millipore Ltd., Cork, Ireland), previously activated in methanol. Total transferred protein was determined by 5 min incubation with Revert^TM^ Total Protein Stain (LI-COR, Lincoln, NE, USA) and the signal was read at 700 nm using the Odyssey Fc Imaging System (LI-COR, Lincoln, NE, USA). Membranes were blocked in Odyssey Blocking Buffer (diluted 1:1 in TBS) (LI-COR) for 1 h at room temperature, and then overnight at 4 °C and in agitation in the corresponding diluted primary antibody. The primary polyclonal antibodies used and corresponding dilutions were: Capn1 (sc-7530; 1/100), Ctsda (sc-6486; 1/200), Ctsl (sc-6501; 1/200), and Mafbx (sc-33782; 1/400) (Santa Cruz, CA, USA). These primary antibodies were previously validated for gilthead sea bream [13,37,47]. The antibody species were Rabbit for Mafbx and goat for the other proteins. After washing with TBS-T, the membranes were incubated with the goat antirabbit fluorescence secondary antibody for Mafbx and donkey antigoat for the other proteins (Cat. No. 925-32211 and Cat. No. 925-32214, respectively, Servicios Hospitalarios) for 1 h at room temperature at a dilution of 1/10,000 for Mafbx and 1/5000 for the other proteins. After incubation, the membranes were imaged at 700 nm (Mafbx) or 800 nm (the other membranes). To reuse the membranes, they were immersed in a commercial stripping buffer (NewBlot PVDF 5X Stripping Buffer, LI-COR, Lincoln, NE, USA) for 20 min at room temperature. From the images obtained, the quantification of the proteins was performed using the Odyssey Software Image Studio v. 5.2.5. (LI-COR, Lincoln, NE, USA). The raw images captured for the Western Blot analysis are compiled in the Appendix A.

### 2.5. Muscle Texture Measurement

Texture analyzes were performed at the Departament d’Enginyeria Agroalimentària i Biotecnologia of the Universitat Politècnica de Catalunya (ESAB, Castelldefels, Spain) using a TA.XT2i Texture Analyzer (Stable Micro Systems Ltd., Godalming, UK) coupled to a Mini Kramer HDP/MKO5 (Stable Micro Systems Ltd., Godalming, UK). As texture analysis, maximal strength and elasticity were measured using the muscle pieces (1.5 × 1.5 cm) extracted from the anterior-dorsal region. Maximal strength is defined as the maximal force applied to cut the sample completely. Elasticity is the capacity of the muscle to recover its initial aspect following the application of force, and it coincides with the linear portion of the texture curve before break point. Both parameters were measured as previously described [36,48].

### 2.6. Statistical Analysis

Data were analyzed using IBM SPSS Statistics v. 25 (IBM Corp., Armonk, NY, USA) and presented as means + standard error of the mean (SEM). A Shapiro–Wilk test was performed to analyze the normality of the data, and homogeneity of the variances was tested with a Levene’s test. Data were analyzed by a two-way analysis of variance (ANOVA) with diet (HP, HE) and swimming activity (VS: voluntary swimming; EX: exercise) set as independent factors, and it was followed by a Tukey’s post-hoc test. Differences among groups were considered significant at *p* < 0.05.

## 3. Results

### 3.1. GH and IGF-1 Plasma Levels

The GH plasma levels as well as the GH/IGF-1 ratio were affected by the diet composition, the swimming activity, and most importantly, by the interaction between both factors. Hence, in fish under voluntary swimming, the circulating GH plasma levels and the GH/IGF-1 ratio were significantly higher in fish fed with HP in comparison to those fed with HE. However, exercise significantly decreased both parameters in HP-fed fish, thus disappearing the differences among both dietary groups. The IGF-1 levels were not affected by any factor (as illustrated in Figure 1).

### 3.2. GH-IGFs Axis Components Gene Expression in Liver 

In liver, only the gene expression of *igf-1c* was significantly affected by the activity and the interaction of the two factors. Thus, under voluntary swimming, *igf-1c* showed apparent reduced expression in fish fed with HE (although not significant); while the sustained exercise caused a significant increase of *igf-1c* in fish fed with HE, equalizing the expression levels in both dietary conditions (as illustrated in Figure 2). The gene expression of *igf-1a*, *igf-1b,* and *igfbp-1a* was also analyzed in liver but remained unaltered by any of the factors (data not shown).

### 3.3. GH-IGFs Axis Components Gene Expression in Anterior and Caudal Muscle

With regards to the white skeletal muscle, in the anterior region, diet significantly modified *igfbp-1* expression; swimming activity altered the *ghrs*, while *igf-2*, *igfbp-1a*, *igf-1rb,* and *ghr-2* showed interaction of both variables. The mRNA levels of *igfbp-1a* and *ghr-2* were significantly higher in fish under voluntary swimming and fed with HP diet when compared to that of those fed with the HE. In the exercise condition, these differences disappeared due to the significant decrease of *igfbp-1a* and *ghr-2* in the HP-fed group (as illustrated in Figure 3). 

In the caudal muscle, the gene expression of the GH-IGFs axis components was similarly affected, responding *igfbp-1a* to diet, *ghr-1* to exercise, and *igf-2* and *ghr-2* to the interaction. Swimming activity significantly reduced *ghr-2* levels in the HP-fed group but not in those fish fed with HE diet (as illustrated in Figure 4). In both muscle regions, the expression of the *igf-1* splice variants was also analyzed but was not affected by any of the factors (data not shown).

### 3.4. Proteolytic Markers Gene and Protein Expression

In the anterior muscle the statistical analysis indicated that diet significantly affected the gene expression of *capn1*, *capn3*, *capsn1a*, *ctsl*, and *mafbx*, whereas swimming activity altered *capn3, capsn1a, ctsl, mafbx, n3 and ub*; thus, showing many of these markers interaction effect between both variables. Fish under voluntary swimming and fed with HE diet showed significantly higher mRNA levels of *capn1*, *capn3,* and *capns1a*, and lower *ctsl* and *mafbx* expression compared to that of fish fed with HP. Nevertheless, in the exercise condition, the gene expression of all the proteolytic systems in this muscle region remained stable regardless of the diet composition. These results could be explained by the significant decrease in HE-fed fish of *capn1*, *capn3,* and *capns1a*, as well as the diminution in HP-fed group of *ctsl* and *mafbx* compared to that of the voluntary swimming condition, thus reflecting the diet, activity and interaction effect. The exercise also reduced the expression of *n3* and *ub* in HE-fed fish (as illustrated in Figure 5). Western blot analysis followed a similar tendency, supporting the gene expression results, since protein expression of Mafbx was significantly downregulated in fish under voluntary swimming and fed with HE diet, while in the exercise condition, these differences were not found. Only the protein expression of Mafbx showed significant diet and activity effects (as illustrated in Figure 6).

With respect to the transcriptional profile of the proteolytic systems in the caudal muscle, similar results were observed to those obtained in the anterior region and diet affected the same genes plus *n3* and *ub.* The activity effect was only observed in *ctsl;* while *capn1*, *capn3*, *caps1a*, *ctsl*, *mafbx*, *n3,* and *ub* showed interaction of both variables. In voluntary swimming, HE diet significantly enhanced the gene expression of *capn1*, *capn3,* and *capns1a* in fish muscle, as in the anterior region, but also that of *n3* and *ub*. Likewise, the expression of *ctsl* and *mafbx* was significantly decreased in fish fed with the HE diet in voluntary swimming. However, in the exercise condition, swimming activity attenuated these differences observed in voluntary swimming, as it provoked a significant decrease of *capns1a* and *ub* in the HE-fed group, an increase of *capn1* and *ub* in fish fed with the HP diet and a reduction of *ctsl* and *mafbx* also in HP-fed fish. In the case of *capn3*, exercise induced an inverse expression pattern, increasing its levels in fish fed with HP diet and decreasing those of fish fed with HE, as observed in anterior muscle for this gene (as illustrated in Figure 7).

### 3.5. Myogenic Regulatory Factors and Growth Inhibitors Gene Expression

The transcriptional profile of the *mrfs* and growth inhibitors (*mstns*) in the anterior and caudal muscle remained unaltered by diet in both voluntary swimming and sustained exercise conditions (as illustrated in Figure 8 and Figure 9), although the expression of the *mstns* showed a tendency to increase in the anterior muscle of HE-fed fish in both swimming conditions (as illustrated in Figure 8). In the caudal muscle, the swimming activity diminished the expression of *mrf4*, *mstn1,* and *mstn2* in both dietary conditions, being clearer the effect in the last two genes in HP-fed fish (as illustrated in Figure 9).

### 3.6. Muscle Texture

Diet composition and activity had significant effects on both parameters of muscle texture, maximal strength, and elasticity, and an interaction effect was also observed on maximal strength. In fish under voluntary swimming, fish fed with HE diet showed lower values of maximal strength and the same trend in elasticity compared to that of those fed with HP diet. Nevertheless, these diet-induced differences disappeared in fish subjected to sustained exercise due to the increase of these parameters in fish fed with HE (as illustrated in Figure 10).

## 4. Discussion

Beneficial effects of exercise were reported in fish, including enhanced muscle growth and feed conversion efficiency [6,8,12,15,49], although the role of the GH-IGFs axis in these effects remains controversial [7]. In addition, it is still necessary to better understand how an energy demanding condition, like forced and sustained exercise, alters nutrients utilization. These insights in exercised fish would allow us to know the maximal grade of inclusion of lipids or carbohydrates in the diet, which allow the reduction of nitrogen discharges and feeding costs without compromising the metabolism, and consequently, growth. The present work is an extension of a previous one in which it was evaluated how nutrient balance affects growth, muscle composition, and mitochondrial metabolism depending on the physical activity conditions [5]. Briefly, under voluntary swimming, fish fed with HE diet showed retarded growth and higher lipid deposition in muscle compared with that of those fed with HP diet, while these differences were not present in the exercised fish. Similarly, the hepatic expression of energy metabolism and mitochondrial biogenesis markers revealed clear differences between dietary groups in nonexercised fish which were not observed with swimming activity [5]. In the current work, we focused on the response of the GH-IGFs axis, proteolytic systems and other muscle developmental markers’ expression depending on diet formulation and physical activity, aiming to provide valuable information for the applicability of exercise for a more sustainable farming of fish in aquaculture.

Many studies demonstrated that changes in fish growth rate are often followed by modifications in the GH and IGF-1 plasma levels, as these are the main muscle-accretion regulatory factors [12,14,16,50]. The results obtained in the voluntary swimming condition revealed that the greater weight gain of HP-fed fish compared to that of HE-fed group was accompanied by significantly higher plasma GH levels, which led to an increased GH/IGF-1 plasma ratio in this group. Dietary proteins and lipids can influence the homeostasis of the somatotropic axis; however, there are some discrepancies in the effects of the proportion of these nutrients on GH secretion in fish [51,52,53]. The protein or lipid content affected differently the plasma GH levels depending on the ration size in gilthead sea bream juveniles [53]. Fish fed *ad libitum* shows upregulated GH secretion by a high-lipid diet; while fish under fixed feeding levels appear to have increased circulating GH when fed a high-protein diet. Hence, our results could suggest that under a 5% meal ration, the higher protein content of HP diet had stronger effects on GH secretion than that of the lipids of HE diet. Furthermore, considering the saturation of the liver’s mitochondrial oxidative systems found in fish fed with HE [5], the lipolytic effects of GH possibly impaired the metabolic condition in this group. In fish subjected to sustained exercise, differences in GH levels due to diet were not observed, as a consequence of a downregulation of GH production by swimming activity in those fish fed with HP diet, as we observed in different studies [12,14]. Moreover, these comparable results on circulating GH and IGF-1 levels are consistent with the similar final body weight that exercised fish showed compared to that of those under voluntary swimming, regardless of diet composition [5], pointing out that exercise caused a diet-dependent differential response of these parameters, which is clearly reflected by the significant interaction between factors.

The hepatic transcriptional profile of the GH-IGFs axis components showed few significant differences between groups. The mRNA levels of *igf-1c* in HE-fed fish showed a significant increase in the group exposed to sustained exercise in comparison to that of the values in voluntary swimming, thus reaching the levels of the HP-fed group. These results would support a possible important implication of this *igf-1* isoform in fish growth regulation, as suggested in a previous study [30]. In fish under voluntary swimming and fed with HE, the *igf-1rb* expression was lower compared to that of fish fed with HP; while in exercise, the activity caused a significant increment of its expression in HE-fed group, thus equalizing the levels of both conditions. Our results only reflect a snapshot from the time of sampling, and growth is a dynamic process in which the gene expression and synthesis of GH-IGFs axis members can experience changes over time [54,55]. In any case, of the similar *igf-1rb* hepatic expression in fish under sustained exercise is congruent with the absence of differences in their final body weight, since the growth-promoting effects of IGFs are mediated through their interaction with IGF-1Rs [17,20].

Regarding the GH-IGFs system expression in muscle, important changes were not observed in any of the experimental groups, neither in the anterior nor in the caudal muscle. The main significant differences observed were in the anterior region of fish in voluntary swimming fed with HP diet, which presented higher expression of *igfbp-1a* and *ghr-2* than those fed with HE. *igfbp-1a* and *ghr-2* are considered negative regulators of growth in fish since their protein and/or gene expression are usually upregulated in catabolic conditions (e.g., hypoxia, stress or fasting) [56,57,58,59]. However, the overexpression of muscle *igfbp-1a* and *ghr-2* that we found in fish fed with HP could indicate a specific condition produced after the period of faster growth showed in this group. In fact, the higher expression of *ghr-2* found is in concordance with the high circulating GH levels in the HP-fed group, and also with the stronger response of *ghr-2* isoform to a nutritional treatment reported by Benedito–Palos et al. [24] in the same species. In gilthead sea bream, the gene expression of *ghrs* in liver and skeletal muscle generally increases along with GH concentration in plasma [56,59]. Hence, the positive response of *ghrs* expression to increased circulating GH might be a mechanism to prevent an excess of GH signaling, since *ghrs* can experience post-transcriptional modifications that generate truncated GHRs without the intracellular signaling domain [56,57,59]. Furthermore, the truncated forms are assumed to be the preferential substrate for proteolytic cleavage to produce the circulating binding proteins (GHBPs) [60,61], which regulate half-life and bioavailability of GH [20]. This scenario of genes expression in voluntary swimming was not observed when juveniles were subjected to sustained exercise, as the exercise reduced in the anterior muscle *igfbp-1a,* and *ghr-2* expression in the HP-fed fish, in agreement with the results of GH and GH/IGF-1 ratio in plasma. The interaction between diet and swimming activity significantly affected *igf-2* expression in the anterior and caudal muscle, but this response was not observed in *igf-1* expression, in agreement with the differential role of both IGFs already found in this species [21]. Summarizing, the changes observed in GH-IGFs system in muscle suggest that exercise tends to equalize muscle growth conditions in both dietary groups.

The transcriptional profile of the proteolytic markers in muscle showed that in voluntary swimming, HE feeding induced a significant upregulation of *capn1*, *capn3*, *capns1a*, *n3,* and *ub* mRNAs in either the anterior or caudal muscle region, while *ctsl* and *mafbx* were downregulated, also at a protein level in the case of Mafbx. Calpains appear to be more involved in the proliferation stages of the myogenesis, as observed in different species, including gilthead sea bream [31,47,62]. Regarding the cathepsins and the UbP system, they seem to have greater importance in myogenic differentiation and formation of myotubes, as suggested in gilthead sea bream [47] and Atlantic salmon (*Salmo salar*) [63]. Moreover, in gilthead sea bream fasted for 21 days, the expression of calpains is rapidly upregulated within 24 h of refeeding, whereas the cathepsins and the UbP system members respond one week later or even do not respond [25]. Altogether, in the present study, the upregulation of the calpains, along with the decrement of *ctsl* and *mafbx* in the voluntary swimming group fed with HE diet, could indicate that the myogenesis in these fish is in a less-advanced stage compared to in those fish fed with HP; in agreement with the lower body weight observed in HE-fed fish [5]. Furthermore, the saturation of the lipid oxidation systems in this group would favor the utilization of amino acids generated by the proteolytic systems [5]. Nevertheless, in fish exposed to sustained exercise, the differences among both diet groups in the proteolytic systems’ expression almost disappeared, basically due to the decreased expression of those genes upregulated in voluntary swimming. This response agrees with the hypothesis that the high-energy demand induced by the sustained exercise generates a metabolic switch that promotes the optimization of nutrients use [5]. Therefore, in exercised fish fed with HE diet, lipid utilization as an energy source was improved, resulting in a protein-sparing effect. These data agree with the similar final body weight observed in exercised fish regardless of diet composition. The regulation of muscle gene and protein expression does not follow an identical pattern [25], as shown here by Capn1, Ctsl, and Ctsda; but it is interesting that in the case of Mafbx both gene and protein expression are increased in HP-fed fish under voluntary swimming, again supporting that this group was in a more advanced myogenic condition. Overall, the different responses of the proteolytic systems to distinct dietary regimes and physical activity observed in the current study open the possibility of using them as markers of nutritional status and culture conditions.

The gene expression of the *mrfs* and *mstns* was not affected by diet in any of the muscle regions of fish in voluntary swimming, suggesting that after six weeks of experiment the effects on myogenesis were still not noticeable. In gilthead sea bream fingerlings exposed to exercise for six weeks, the myogenic factors are slightly affected in the anterior muscle, while the proteolytic genes appear to be already upregulated to start muscle reorganization [13]. This response agrees with the important changes in proteolytic genes expression found in this study in fish under voluntary swimming and probably the induction of myogenesis will follow the activation of the proteolytic genes, as the importance of the proteolytic systems to facilitate the recovery of the mature muscle fiber by activating the myogenic program was reviewed [32]. Hence, it is understandable that the main genes involved in the regulation of muscle development are not still showing significant responses after 6 weeks of treatment. In fish under exercise, myogenic genes expression presented a similar pattern to that of fish in voluntary swimming, showing no significant differences in any of the muscle regions at the time analyzed. Nevertheless, it is remarkable that only in the caudal muscle the *mstns* expression was downregulated by exercise compared to levels in voluntary swimming, significantly in fish fed HP. This decrement of *mstn1* and *mstn2* would suggest a lower growth repression in this muscle region of fish subjected to sustained exercise since Mstns are the main inhibitors of muscle development in vertebrates [40,41]. These results are in concordance with the different responses to exercise along the muscle trunk reported in gilthead sea bream [6,13].

Muscle texture was analyzed by the measurement of maximal strength and elasticity which are two well established parameters to determine the physical properties of the flesh. In general terms, fish fillets with certain degree of firmness and elasticity are preferred, while low values of these two factors are associated with product defects. The results of maximal strength and elasticity showed the same tendency as that observed with the other variables studied. In fish under voluntary swimming and fed with HP diet, both parameters followed a similar trend, being significantly higher in maximal strength in comparison with values of HE-fed fish. However, exercise in fish fed with HE caused an increase on muscle texture, reaching the levels of the HP-fed group. These results agree with the gene expression data of proteolytic markers, which in exercise condition do not present the differences observed in the fish under voluntary swimming. These findings could indicate a negative relationship between calpains’ mRNA levels and flesh firmness, as fish fed with the HE diet in voluntary swimming showed higher expression of these proteases and lower texture. Similar results were described also in gilthead sea bream under fasting and refeeding conditions, thus suggesting a valuable applicability of the calpains as markers for flesh quality analysis [37]. Our results were consistent with previous reports on negative correlation between muscle texture and fat content [64,65]. Fish under voluntary swimming and fed with the HE diet presented significantly higher lipid deposition in muscle [5] and lower values of maximal strength and elasticity, but these differences were not found in the exercise condition. Altogether, these muscle texture results demonstrated the positive and interesting effects of sustained swimming to compensate or ameliorate the reduced texture parameters that are associated with high-fat feeding in fish farming. 

## 5. Conclusions

The results obtained in this study provide convincing evidences that sustained exercise may compensate nutrient imbalances provoked by high-fat diets, improving fish growth performance and flesh texture. Furthermore, the response of the proteolytic systems to the composition of the diet and physical activity proposes them as valuable markers of fish nutritional status and muscle growth.

## Figures and Tables

**Figure 1 animals-11-02182-f001:**
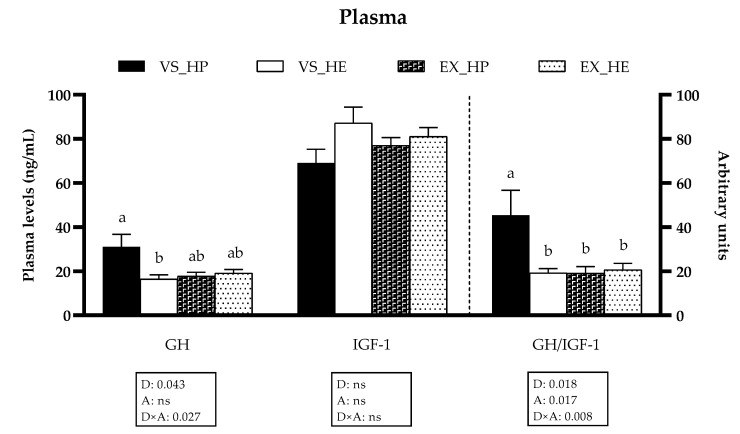
Growth hormone (GH) and insulin-like growth factor-1 (IGF-1) plasma levels of fish under voluntary swimming (VS) or sustained exercise (EX) and fed with high-protein diet (HP) or high-energy diet (HE). Data are shown as means + SEM. VS, *n* = 10; EX, *n* = 22. Factorial statistical analysis was assessed by two-way ANOVA, and *p*-values of factors diet (D), physical activity (A) and interaction (D×) are displayed under graph. Different letters indicate significant differences (Tukey’s post-hoc test, *p* < 0.05).

**Figure 2 animals-11-02182-f002:**
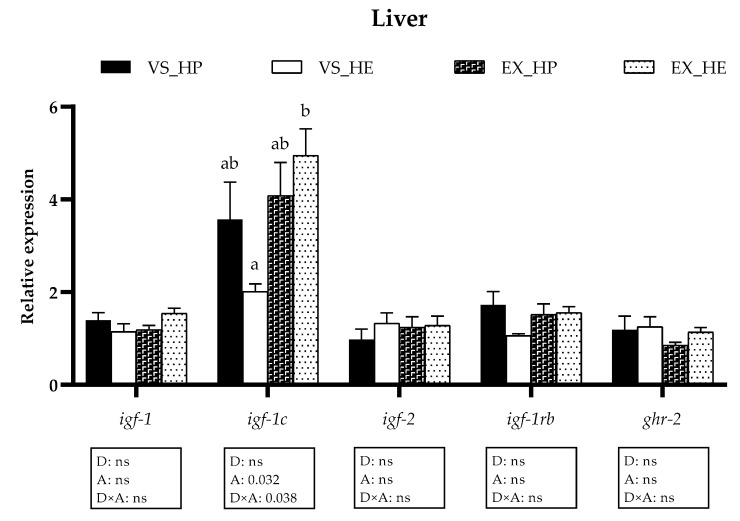
Relative gene expression of GH-IGFs axis members in liver of fish under voluntary swimming (VS) or sustained exercise (EX) and fed with the high-protein diet (HP) or high-energy diet (HE). Data are shown as means + SEM. VS, *n* = 12; EX, *n* = 16. Factorial statistical analysis was assessed by two-way ANOVA, and the *p*-values of factors diet (D), physical activity (A) and interaction (D×A) are displayed under graph. Different letters indicate significant differences (Tukey’s post-hoc test, *p* < 0.05).

**Figure 3 animals-11-02182-f003:**
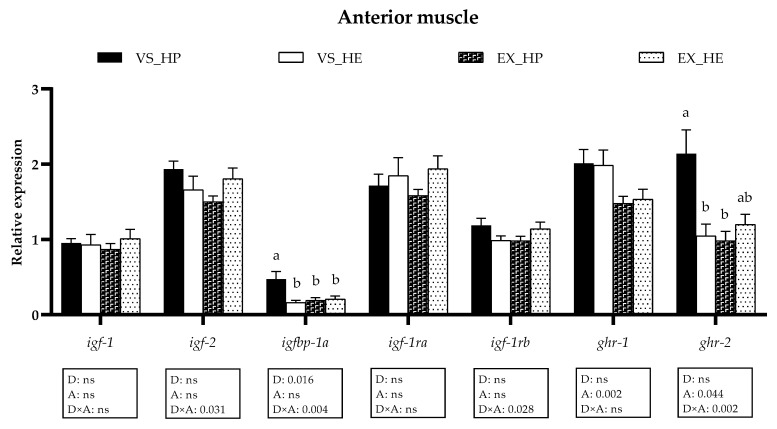
Relative gene expression of the GH-IGFs axis members in anterior muscle of fish under voluntary swimming (VS) or sustained exercise (EX) and fed with the high-protein diet (HP) or high-energy diet (HE). Data are shown as means + SEM. VS, *n* = 12; EX, *n* = 16. Factorial statistical analysis was assessed by two-way ANOVA, and *p*-values of factors diet (D), physical activity (A), and interaction (D×A) are displayed under the graph. Different letters indicate significant differences (Tukey’s post-hoc test, *p* < 0.05).

**Figure 4 animals-11-02182-f004:**
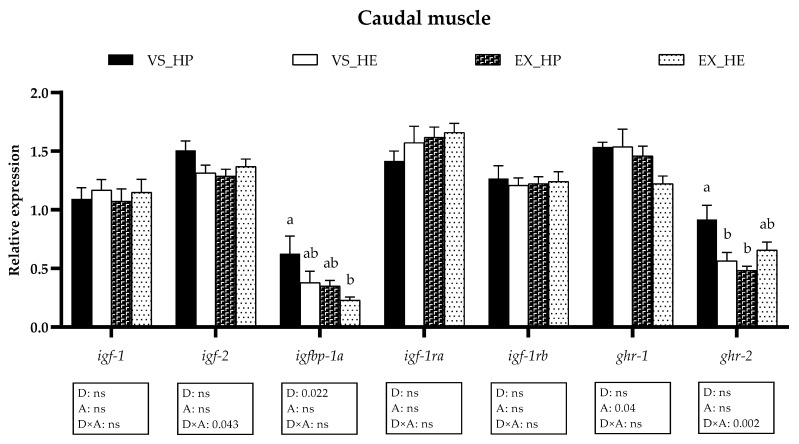
Relative gene expression of the GH-IGFs axis members in caudal muscle of fish under voluntary swimming (VS) or sustained exercise (EX) and fed with high-protein diet (HP) or high-energy diet (HE). Data are shown as means + SEM. VS, *n* = 12; EX, *n* = 16. Factorial statistical analysis was assessed by two-way ANOVA and *p*-values of factors diet (D), physical activity (A), and interaction (D×A) are displayed under the graph. Different letters indicate significant differences (Tukey’s post-hoc test, *p* < 0.05).

**Figure 5 animals-11-02182-f005:**
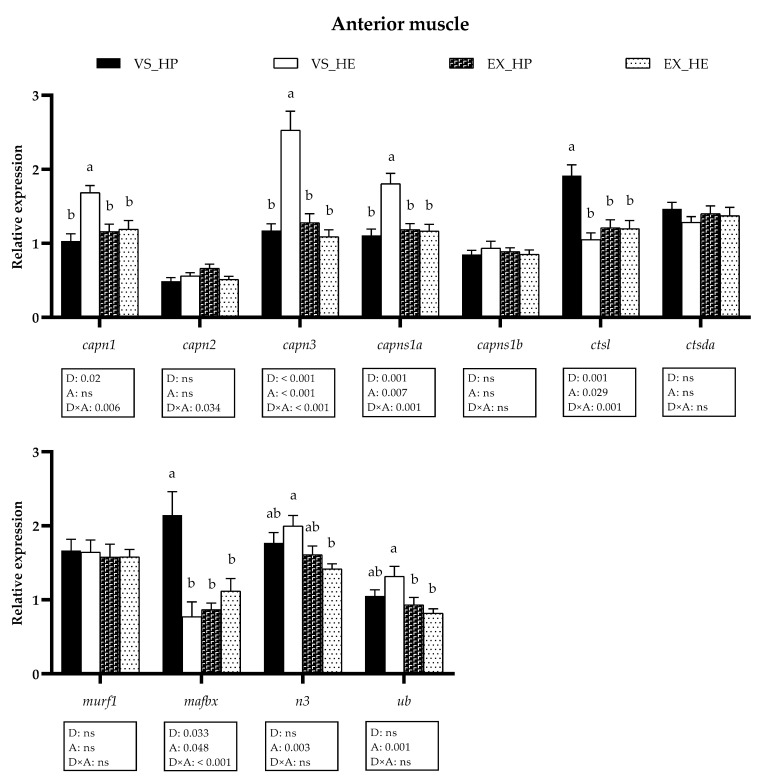
Relative gene expression of proteolytic markers in anterior muscle of fish under voluntary swimming (VS) or sustained exercise (EX) and fed with high-protein diet (HP) or high-energy diet (HE). Data are shown as means + SEM. VS, *n* = 12; EX, *n* = 16. Factorial statistical analysis was assessed by two-way ANOVA and *p*-values of factors diet (D), physical activity (A), and interaction (D×A) are displayed under graph. Different letters indicate significant differences (Tukey’s post-hoc test, *p* < 0.05).

**Figure 6 animals-11-02182-f006:**
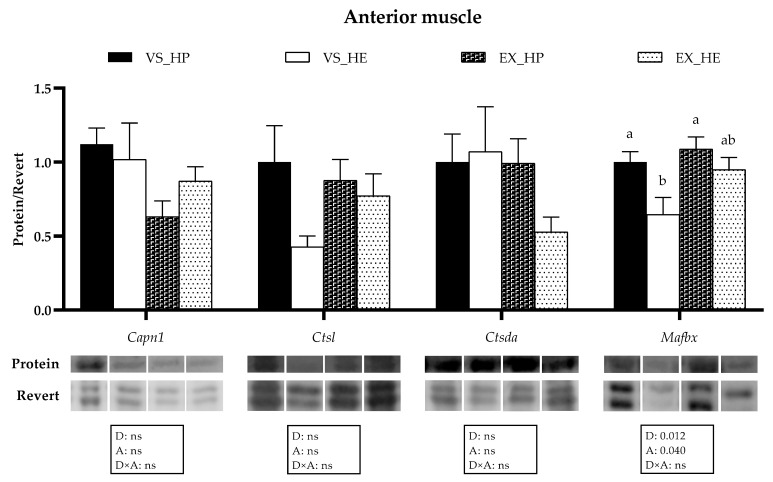
Representative western blot and densitometric analysis of proteolytic markers’ protein levels in anterior muscle of fish under voluntary swimming (VS) or sustained exercise (EX) and fed with high-protein diet (HP) or high-energy diet (HE). Bands were normalized to their Revert^TM^ total protein staining (corresponding well is shown). To eliminate intermembrane variability, relative intensity of each specific band was normalized by geometric mean of intensity of that of VS_HP diet group of corresponding membrane. Data are shown as means + SEM. VS and EX, *n* = 6. Factorial statistical analysis was assessed by two-way ANOVA, and *p*-values of the factors diet (D), physical activity (A), and the interaction (D×A) are displayed under graph. Different letters indicate significant differences (Tukey’s post-hoc test, *p* < 0.05). (The raw images captured for the Western Blot analysis are compiled in the Appendix A).

**Figure 7 animals-11-02182-f007:**
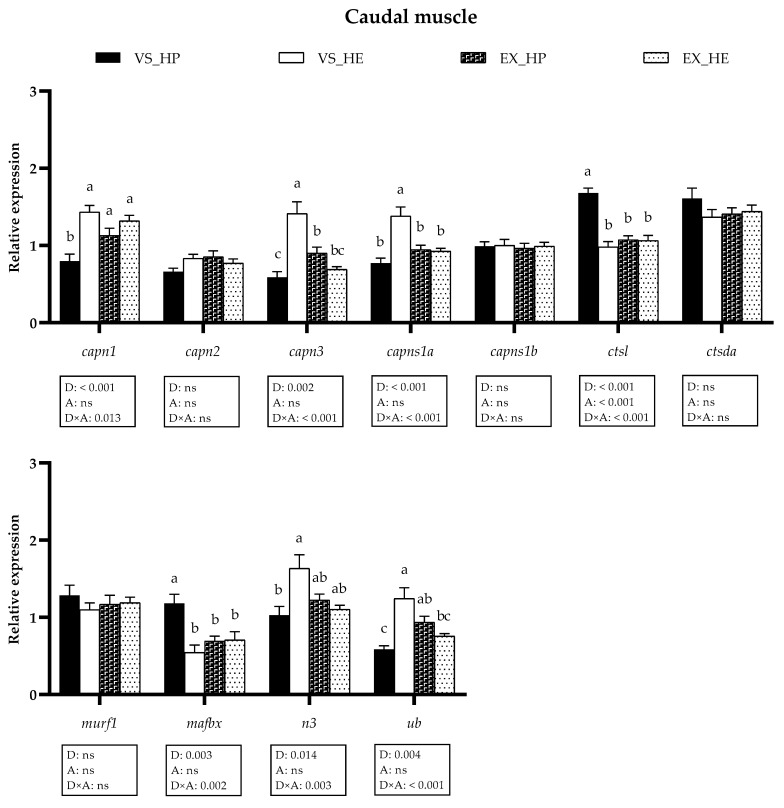
Relative gene expression of proteolytic markers in caudal muscle of fish under voluntary swimming (VS) or sustained exercise (EX) and fed with high-protein diet (HP) or high-energy diet (HE). Data are shown as means + SEM. VS, *n* = 12; EX, *n* = 16. Factorial statistical analysis was assessed by two-way ANOVA, and *p*-values of the factors diet (D), physical activity (A), and interaction (D×A) are displayed under the graph. Different letters indicate significant differences (Tukey’s post-hoc test, *p* < 0.05).

**Figure 8 animals-11-02182-f008:**
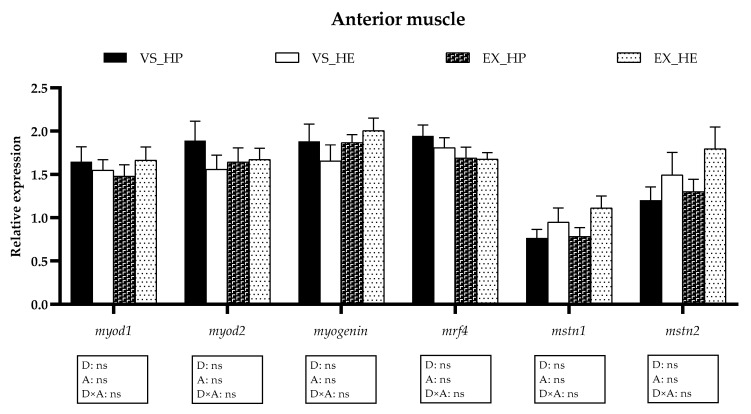
Relative gene expression of myogenic regulatory factors in anterior muscle of fish under voluntary swimming (VS) or sustained exercise (EX) and fed with the high-protein diet (HP) or high-energy diet (HE). Data are shown as means + SEM. VS, *n* = 12; EX, *n* = 16. Factorial statistical analysis was assessed by two-way ANOVA and *p*-values of factors diet (D), physical activity (A), and interaction (D×A) are displayed under graph.

**Figure 9 animals-11-02182-f009:**
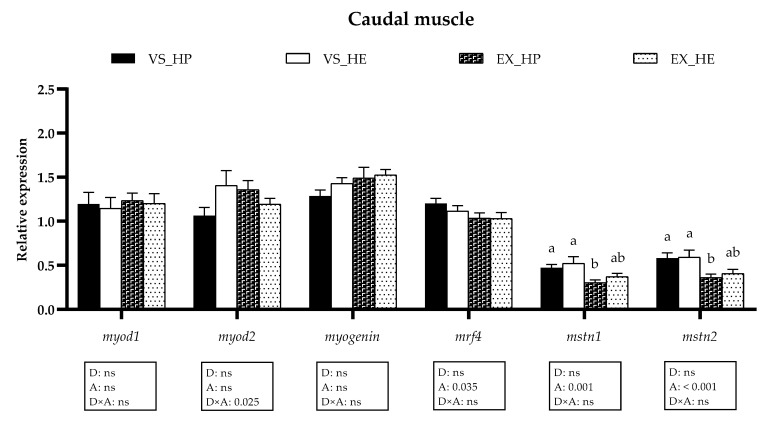
Relative gene expression of myogenic regulatory factors in caudal muscle of fish under voluntary swimming (VS) or sustained exercise (EX) and fed with high-protein diet (HP) or high-energy diet (HE). Data are shown as means + SEM. VS, *n* = 12; EX, *n* = 16. Factorial statistical analysis was assessed by two-way ANOVA and *p*-values of factors diet (D), physical activity (A), and interaction (D×A) are displayed under the graph. Different letters indicate significant differences (Tukey’s post-hoc test, *p* < 0.05).

**Figure 10 animals-11-02182-f010:**
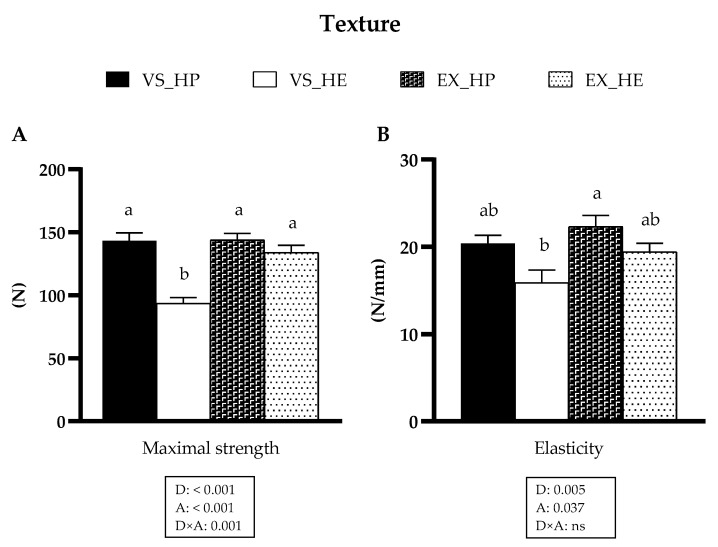
Muscle maximal strength (**A**) and elasticity (**B**) of fish under voluntary swimming (VS) or sustained exercise (EX) and fed with high protein diet (HP) or high-energy diet (HE). Data are shown as means + SEM. VS, *n* = 10; EX, *n* = 16. Factorial statistical analysis was assessed by two-way ANOVA, and *p*-values of factors diet (D), physical activity (A), and interaction (D×A) are displayed under graph. Different letters indicate significant differences (Tukey’s post-hoc test, *p* < 0.05).

**Table 1 animals-11-02182-t001:** Composition of diets used in experimental trial.

Item	HP Diet	HE Diet
Digestible energy (MJ/kg)	18	19.9
Protein (% dry mass)	54	50
Lipids (% dry mass)	15	20
DHA (% dry mass)	15	20
EPA (% dry mass)	1	1.4
ARA (% dry mass)	2.5	3
DHA/EPA/ARA	5/12.5/1	3.5/7.5/1
Cellulose (% dry mass)	1.6	1.7
Ashes (% dry mass)	10.5	6.4
Total P (% dry mass)	1.4	1.2
Estimated nitrogen-free extract	20.5	23.5

MJ: megajoules; DHA: docosahexaenoic acid; EPA: eicosapentaenoic acid; ARA: arachidonic acid; P: elemental phosphorus.

## Data Availability

The data presented in this study are available in the current article and its corresponding Appendix A.

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
