# Peer review of "Diet and Exercise Modulate GH-IGFs Axis, Proteolytic Markers and Myogenic Regulatory Factors in Juveniles of Gilthead Sea Bream (*Sparus aurata*)"

_animals, 2021, doi:10.3390/ani11082182_

Round 1

Reviewer 1 Report

This is a very interesting and well conducted study that investigates the combined effects of moderate exercise and diet on growth performance in gilthead sea bream juveniles. Results also suggest that that exercise might be a useful tool to minimize nutrient imbalances, and that proteolytic genes could be good markers of the culture conditions and dietary treatments in this species. It is also clear to see that a lot of hard work has been put into the study. This is a welcome manuscript for the journal and I do not find any negative points in the paper.

Author Response

This is a very interesting and well conducted study that investigates the combined effects of moderate exercise and diet on growth performance in gilthead sea bream juveniles. Results also suggest that that exercise might be a useful tool to minimize nutrient imbalances, and that proteolytic genes could be good markers of the culture conditions and dietary treatments in this species. It is also clear to see that a lot of hard work has been put into the study. This is a welcome manuscript for the journal and I do not find any negative points in the paper.

We fully appreciate your supporting comments to our manuscript.

Reviewer 2 Report

This study is the follow-up of a previous experiment by the same group, wit the aim here at looking more in detail at the effect of diet and exercise on the GH-IGFs axis of juvenile seabream. The experiment is well designed and the conclusions sound.

My two points of concern are the statistical analyses, the authors chose to perform a 2-way ANOVA (right choise), however they should be followed by Tukey tests, not student t-tests (at least not uncorrected), and only if there is a significant effect detected by the two-way ANOVA, please have another look at the statistical analysis and correct what is necessary.

My second point concern the use of 'sustainable', which need to be developed, fat or carbohydrates as replacement for protein are not necessarily more sustainable, it is a bit more complex (see specific comments below), it depends on a lot of factors and the authors need to take this into account to avoid misleading the reader.

General remark:

Please change ‘Gh’ and ‘Igf’ to ‘GH’ and ‘IGF’ respectively to match standard abbreviations

Simple summary

L26 please rephrase ‘and decreased muscle texture respect fish fed….’

L24 and 27, and elsewhere, please harmonize the terminology used to describe the activity (forced swimming L24, moderate exercise L27, exercise L37, sustained and moderate exercise L41 ect)

L31 (and also in the introduction), L59-64. The term ‘more sustainable’ here can be misleading and is not necessarily correct. It is true that (fish) protein in fishmeal are expensive and not sustainable, however, fish oil is also expensive and not sustainable as well, the HE diet contains 4% less protein but 5% more fat. It is not mentioned in the manuscript the origin of the protein nor the oil contained in the two diets (please add it in the material and method). Secondly, the authors should also think in term of circular economy and life cycle analyses as well when talking about the sustainability of an ingredient. Lipid or carbohydrate could also be not the best alternatives if for instance they come from the fisheries industry (oil) or if they come from land-based industries with an ecological impact or potentially competing with the human food industry (palm oil). There are good, sustainable, alternatives to conventional fish proteins already existing or being under development (valorization of by-products from several industries). Those aspects should be considered by the authors.

Abstract

L34, please don’t use abbreviations (HP/HE) without mentioning what they mean

L37 (‘but in exercise’) and L38 (‘was little’) affected. Please check the English/rephrase

L42. Delete ‘those’

Keywords

Perhaps consider adding additional keywords (Sparus aurata? Aquaculture? Swimming (exercise)?)

Introduction

L49-50. Maybe this sentence is a bit a shortcut. Perhaps consider developing more: 1) The population is growing worldwide; we need more food. 2) Fish are generally considered a healthy source of protein, fat and vitamin, the aquaculture sector is growing and can help face this demand 4) However, the sector is facing problems such as…

L50-54. Please add some reference for those statement. Please also mention more clearly the ecological impact in general caused by the use of fish proteins and oil in fish meal (not only increase of N-load); overfishing/destruction of habitat ect.

L54-54, please add the reference of your studies here. Also, other groups have been investigation this topic as well, I believe it is also worth and fair to mention that (i.e Arjan Palstra)

L61-64. Please make it clearer that this study is the follow-up of your previous work [10]

Material and Methods

Experimental design. Please add some specification about the system (flow-through? RAS?, and add other environmental parameters (salinity, N-X waste…) here or in the supplementary material

L120. Please make the ‘3’ from ‘m3’ superscript.

L120-122. Why did the author use different tanks size for the free swimming vs the forced swimming? Why did the author only have one replicate for the free swimming for each diet (1 400L vs 4 200 L for the forced swimming: 1 replicate vs 4)

L126-128 and 130-133: please add the reference of the diets (including name and pellet size, not only brand)

L144. You can remove ‘Styrofoam boxes under ice’ and just mention they were kept on ice

L165, 183, 205 and maybe elsewhere, please remove the ‘-‘ under the ‘°’ when mentioning temperature

L207, maybe remove ‘Tullagreen’ or ‘Cork’

L211, please correct ‘Odissey’ by ‘Odyssey’

L216-225, please remove the capital letters when mentioning animals used for the anti-bodies

Statistical analyses (L234-244)

L238, please remove ‘NY’

L242, please mention in full what VS and EX are, it is the first time you use those abbreviations.

Please revise your statistical methodologies. It is not correct to use T-test after 2-way ANOVA without correction. Please use Tukey test. Moreover, when the two way ANOVA does not show statistical differences, it is not correct to test for differences using T-tests (ex: fig , L284: for igf-1rb  and ghr-2, there is ns for D, A and DxA, so there is no basis to conduct a T-test.

Results

L247-254. This should better be placed in the introduction, shortened

L256, please replace ‘clearly’

Figure 1, L264. Please replace ‘ml’ by ‘mL’ (like in 159)

Figures:

In general, after checking the statistics, please change the figures to make it more clear and homogeneous, you use a square bracket with a star in the middle when you want to highlight differences between diets, and a cross on top of one bar when showing differences between swimming activity. Please also use brackets and place the symbol for difference in the middle as well (see illustration).  

Figure legend: You mention a different N than in the material and method (ex, Fig 1, VS: 10; EX: 22, L135: 12 fish for each VS tanks, 16 for each diet, please indicate how many fish for each condition were used instead and why not all the samples where analyzed? Here or in the material and methods

L275-246. As it is NS, I would not mention it

L279, replace ‘trained’ (harmonize with earlier)

L280, please rephrase (‘respect to’)

L353, Figure 6. Perhaps consider splitting this figure into two sub-parts; Fig 6A and 6B. Then adapt the legend and the text

L413, perhaps mention those two parameters

L419, Figure 10, please split this figure into Figure 10A (maximal strength) and 10B elasticity.

Please also mention what N in ‘(N)’ and ‘(N/mm)’ refers to

Discussion

L475-476. If the P>0,05, you cannot say it was downregulated. Rephrase/delete

L483. Please double check with the journal’s guideline if ‘in vitro’ should be in italic or not

L526-533. Please check if ‘calpains’ and ‘cathepsins’ should be in italic L 526 and L528, as they are L527 and 533 respectively. I have also some doubt with ‘Capn1’, ‘Ctsda’, ‘Ctsl’ abd ‘Mafbx’ L550-551 and ‘Mstns L573

L538. Please quote ‘[10]’ after ‘in HE-fed fish’

L576-578, please expand a bit why the maximal strength and elasticity are good markers of flesh quality. Also, perhaps put your findings a bit into perspective, you are analyzing parameters on juvenile fish of few grams, things can be quite different in adults. Perhaps discuss this perspective?

L587-591, please rephrase; turn it around.

Conclusion

L598, Please rephrase ‘thus causing’

Reference

L650-651, reference [10], please mention the date of acceptance of your other study (21-05-201)

Author Response

We appreciate the referee's comments. Please find the cover letter attached.
